# Development of an Artificial Neural Network Algorithm Embedded in an On-Site Sensor for Water Level Forecasting

**DOI:** 10.3390/s22218532

**Published:** 2022-11-05

**Authors:** Cheng-Han Liu, Tsun-Hua Yang, Obaja Triputera Wijaya

**Affiliations:** 1Department of Civil Engineering, National Yang Ming Chiao Tung University, Hsinchu 30010, Taiwan; 2Department of Civil Engineering, Parahyangan Catholic University, Bandung 40141, Indonesia

**Keywords:** edge computing, ANN, microprocessor, water level prediction, decentralized

## Abstract

Extreme weather events cause stream overflow and lead to urban inundation. In this study, a decentralized flood monitoring system is proposed to provide water level predictions in streams three hours ahead. The customized sensor in the system measures the water levels and implements edge computing to produce future water levels. It is very different from traditional centralized monitoring systems and considered an innovation in the field. In edge computing, traditional physics-based algorithms are not computationally efficient if microprocessors are used in sensors. A correlation analysis was performed to identify key factors that influence the variations in the water level forecasts. For example, the second-order difference in the water level is considered to represent the acceleration or deacceleration of a water level rise. According to different input factors, three artificial neural network (ANN) models were developed. Four streams or canals were selected to test and evaluate the performance of the models. One case was used for model training and testing, and the others were used for model validation. The results demonstrated that the ANN model with the second-order water level difference as an input factor outperformed the other ANN models in terms of RMSE. The customized microprocessor-based sensor with an embedded ANN algorithm can be adopted to improve edge computing capabilities and support emergency response and decision making.

## 1. Introduction

The Emergency Event Database (EM-DAT) includes records for 432 disastrous events related to natural hazards worldwide in 2021. Floods dominated these events, with 223 occurrences, with an average of 163 annual flood occurrences recorded in the 2001–2020 period [1]. The losses of life and property caused by floods are tremendous [2]. Structural and nonstructural measures have been devised to prevent or mitigate loss of life and property [3]. The development of early warning systems, which are nonstructural measures, was cited as a critical defense against floods [4]. These systems involve on-site facilities such as bubble gauges, float gauges and pressure sensors, which are installed to observe water level changes; then, these observations are used as indicators to assess the flood potential [5]. Accurate and cost-efficient water level monitoring sensors are required and must be deployed with very high intensity for detailed flood records [6,7,8,9]. Nevertheless, these sensors are only used for water level monitoring, and this kind of application provides limited lead time for decision makers to take response measures to mitigate the impact of disasters [7,8,9,10,11].

Edge computing is a distributed computing paradigm in which computations are largely or completely performed at distributed device nodes known as smart devices or edge devices, as opposed to computations in a centralized cloud environment [12]. By implementing edge computing on sensors, issues prevalent in centralized cloud systems, such as latency and network connection dependency, can be avoided [13]. Since system failure or other misinformation issues occur during the transmission process, which is common during disasters, simulation cannot be performed, resulting in delays in the emergency response and increasing the possibility of damage. Many studies in the fields such as medical care, manufacturing, and fault detection have developed sensors not only for monitoring but also implemented edge computing for applications that are close in proximity to the sources in case of need, e.g., [13,14,15]. In comparison with other applications, only a few studies have investigated edge computing in landslide- and flood-related applications [16,17,18]. People hesitate to apply edge computing in flood warning systems for the following possible reasons. First, most of the systems are still built based on traditional frameworks. If a prediction is performed in real time for operational purposes, the observed data from local monitoring devices are transmitted to the remote server through the internet. The server performs simulations and provides results to responders for decision making. This is called a centralized simulation approach. The monitoring devices are only capable of observing water levels without forecast functions. Therefore, there is no opportunity to implement edge computing until new sensors are developed and deployed. Second, to increase the lead time of the response, physically based models are used in the traditional framework for predictions. These systems consider different hydrological and inundation modeling components based on the specific target region, size of basins, available data and resources, and system development approach [19]. These physically based models considering detailed hydraulic processes (e.g., solving Saint-Venant equations) are complex and computationally intensive, causing limited applications in practical applications due to the availability of input data for parameterization and the detailed requirements of simulations [20,21,22]. None of these models are designed to be installed in sensors with limited computing power. In this regard, data-driven machine models that focus on the relationship between input factors (e.g., historical flow discharge and precipitation) and outputs (e.g., water level) are recommended as an alternative [23,24].

Related studies about the integration of customized sensors and data-driven models were examined. Customized ultrasonic sensors have been developed [5,10]. Warnings are issued when the monitored water levels are above a specific threshold. No extra calculation was performed on the sensors in these studies. Other studies used microcontroller-based sensors to collect environmental information and perform calculations in cloud-based neural networks to predict flood disaster conditions [25,26,27]. These studies confirmed the functionality of ANN models for water level predictions. However, the models were executed in the cloud-based server. Finally, Samikwa et al. [28] utilized edge computing for flood prediction and carried it on a low-power device within the IoT wireless sensor network. Long short-term memory (LSTM), a type of ANN, was applied in the study to predict one-hour ahead-of-time forecasts of water level. Similar to Bande and Shete [25], water level and rainfall were utilized as inputs in the study to train the ANN models to produce forecasts. However, the details of the input selection were not discussed in the study. Al Qundus et al. [29] deployed sensors to collect data such as water levels, temperature and wind speed. A support vector machine (SVM) algorithm was implemented at nodes (sensors). Only four features (temperature, humidity, wind speed, and water level) were selected to develop the SVM models. However, the details of feature selection were not discussed in the study.

This study proposed a novel decentralized flood warning system with edge computing. It consists of edge computing-enabled wireless water level monitoring sensors and a suitable forecasting algorithm that was embedded in these sensors. The newly developed sensor shifts applications, data, and computing power (services) away from centralized points to the logical extremes of a network. In other words, the sensor involves applications or general functionalities that are close in proximity to the sources of other processes, thus involving interactions between distributed systems technology and the physical world; this implies that the sensor can perform simulations and predictions directly at flood-prone locations with localized information. Therefore, situational and customized awareness is maintained during flooding, even if an internet connection is unavailable. As a result, the efficiency of the emergency response is increased. In addition, new flood prediction models are desperately needed to be performed on the computing power-limited sensors. ANN models were developed to determine the water level on the sensor. A detailed analysis regarding the correlation between input factors and output water levels was carefully conducted to maximize the efficiency of edge computing. Furthermore, special attention was given to extreme events such as typhoons during the development of the proposed system.

## 2. Study Areas

This study focused on three rivers and one artificial canal to develop the ANN model and evaluate the performance of the proposed system. The geographical locations of these study areas are shown in Figure 1.

The Yilan River Basin is located on northeastern Taiwan Island. Its main stream is approximately 24.4 km (km) long and covers an area of approximately 149.06 square km. The Yilan River Basin was first selected to train and test the ANN model for water level forecasting. This is because the precipitation, river stage, and flow velocity data have been carefully measured by the Water Resources Agency (WRA) and National Center for High-performance Computing (NCHC) since 2012 [30] for the Yilan River basin. In addition, there is no human interference, such as a reservoir upstream of the Yilan River, and there are no human operation-related factors considered among the ANN input factors in this study. Figure 1 (in a clockwise direction) shows the remaining two river basins, the Beinan River and Toucian River Basins, in eastern and western Taiwan and one artificial canal, Shimen Canal, in western Taiwan. The Beinan River is approximately 84 km long and flows through Taitung County to the Pacific Ocean. The Toucian River flows through Hsinchu County for 63 km to the west. Different from the Yilan River and Beinan River situations, there are also no human-made hydraulic structures upstream along the Toucian River, but there is an off-stream reservoir upstream of the river. These two rivers were selected because there was no human interference, such as reservoirs or gates, found in the rivers. For reference, the proposed system (integration of Raspberry Pi (RPi) sensors and an ANN model) was only implemented on-site in the Shimen Canal because of the limitations of devices and the need for permission to install equipment. Tests performed in the Beinan River and Toucian River were conducted offline.

## 3. System Development

In this study, a water level forecasting model is embedded in an RPi-based monitoring device that can provide real-time water level observations and support local calculations. The system structure and data processing flowchart are illustrated in Figure 2. The monitoring devices use ultrasonic waves to measure the water levels, and the observed data and other related information are preprocessed in the RPi platform. Thereafter, the water level predictions at specific locations are performed using the proposed ANN-based water level forecasting models. The details of each component in the structure are described in the following subsections.

### 3.1. Raspberry Pi Water Level Sensor

Automatic water level sensors with wireless functions are usually costly. Therefore, low-cost, open-source, and low-energy-consumption sensors are always of interest for environmental monitoring [31]. The proposed sensor is shown in Figure 3.

The RPi is a reliable, low-cost microcomputer (MCU) that was developed in 2006 by the University of Cambridge’s Computer Department and has been produced by the Raspberry Pi Foundation since 2012. The RPi 3 Model B+ module, which was released in February 2018 with a 1.4 GHz 64-bit quad core processor, was used as a platform embedded with an ultrasonic sensor to measure water stages at a local site. The UNIX/Debian=based Raspian operating system supports the implementation of a Python programming language-based ANN module to forecast water levels with lead times. Many studies have successfully applied ultrasonic waves to measure water levels under severe environmental conditions [32,33]. The ultrasonic sensor used here is the high-performance ultrasonic distance sensor MB7386 HRXL-MaxSonar-WR from MaxBotix, Brainerd, MN, USA. The ultrasonic sensor emits sound waves at a frequency of 42 kHz with a 6 Hz sampling rate and detects the sound waves that bounce back. The distance can be estimated by the elapsed time between the generated and returning sound waves. Theoretically, the sensor is effective up to a maximum range of 10 m, with functions of temperature compensation and noise cancelation. However, the efficient measuring distance varies based on the size of the target and power supply to the sensor (usually within 5 m).

### 3.2. Artificial Neural Network Water Level Forecasting Algorithm

ANNs are inspired by the human central nervous system. ANNs usually consist of layers such as an input layer, one or more hidden layers, and an output layer. A three-layer ANN data processing flowchart is shown in Figure 4. The input layer comprises a number of nodes (*i* = 1, 2, 3…*n*). A node, also called an artificial neuron, connects to other nodes in the hidden layer and has an associated weight (*w*) and threshold (*bias*). When the incoming signals (*X_1_*, *X_2_*_,_
*X_3_*…*X_n_*) are passed to the nodes (*j* = 1, 2, 3…*N*) in the next layer, they are multiplied by the weight of the connection. These weights describe the importance of any incoming signal, with larger weights contributing more significantly to the output compared to other signals. The effective signal (*E_j_*) to node *j* shown in Equation (1) is the weighted sum of all incoming signals. In the first phase of training, the weights (i.e., *w_ji_*) are set to random values.
(1)$$E_{j}=\overset{n}{\underset{i=1}{\sum }}X_{i}w_{ji}+bias$$

An activation function is used to transform the effective signal (*E_j_*) into an output value to be fed to the next layer or as an output. In this study, hyperbolic tangent (tanh) and exponential linear unit (ELU) functions are used as the activation functions to transfer input signals to hidden and output layers, respectively. The tanh function has an S-shape similar to that of the sigmoid activation function, with a difference in the output range of −1 to 1. The ELU function is also similar to the rectified linear unit (ReLU), with a difference in output value for negative values of input. A three-layer network structure with one input layer, one hidden layer, and one output layer is adopted because of the limited computing power of the RPi 3 Model B+ module in the sensor [27]. No general guidelines exist for specifying the optimal number of nodes required in the hidden layer [34]. The number of nodes in the hidden layers can be estimated using Equation (2), as recommended by Fletcher and Goss [35]. The formula was confirmed by Huang and Foo [36] to provide the optimal network size, resulting in minimum error and a high correlation in the validation data set. There are three output nodes representing the forecasted water levels with lead periods of 1, 2, and 3 h.
(2)N=2n
where *N* is the number of nodes in the hidden layer and *n* is the number of incoming signals. The output from the neural network is calculated by propagating an input signal through each layer until the output layer outputs its values. It is a so-called feed-forward network. As mentioned above, the weights initially are random values and modified by reducing the differences between the output and a known output. The procedure repeatedly optimizes the weights until the value of the objective root mean square error (*RMSE*) function, shown in Equation (3), falls below a certain threshold. In this study, the threshold is 0.01.
(3)$$RMSE=\sqrt{\frac{1}{K}\sum_{i=1}^{K=3}{\left(P_{i}-O_{i}\right)}^{2}}$$
where *P_i_* is the predicted water level and *O_i_* is the observed water level. *K* is the number of outputs and here refers to the forecasted water levels with three lead times. This learning procedure is called the backpropagation approach and was proposed by Rumelhart et al. [37]. It is a method for training the weights in a multilayer feed-forward neural network structure. A Python module Scikit-learn was applied for the ANN model computation [38].

Since the output of the developed ANN model is the water level, the selection of the input factors must be based on the characteristics and shifts of the outputs at a given location. The change in the time series water level is not only dependent on rainfall records in upstream watersheds and river inflows but also related to previous water levels based on river discharge conditions. Moreover, the length of the lag phase of each input factor is influenced by the distance between input and output locations, and it determines the length of the input sequence. In fact, the selection of input factors has a large impact on the accuracy and efficiency of ANN models. There is no global way to select the input factors for an ANN model [39]. Thus, parameter selection for edge computing is important to maximize the computing efficiency of ANNs. The efficiency of calculations must be optimized for the appropriate number of input factors. In this study, a cross-correlation analysis (*cc*^2^), shown in Equation (4), one of the most widely used methods for factor selection, as discussed by Babel and Shinde [39], was carried out between the outputs (forecasted water levels at a given location) and input factors with a lag phase length.
(4)$$cc^{2}=\frac{\sum_{i=1}^{n-k}\left(X_{i}-\bar{X}\right)\left(Y_{i+k}-\bar{Y}\right)}{\sqrt{\sum_{i=1}^{n}{\left(X_{i}-\bar{X}\right)}^{2}}\sqrt{\sum_{i=1}^{n}{\left(Y-\bar{Y}\right)}^{2}}}$$
where *n* is the total number of time sequences in hours and *k* represents the time lag value. X and Y are the water level and a possible input factor, respectively. $\bar{X}$ and $\bar{Y}$ denote average values. Only the factors with a relatively high correlation value with the output were adopted in the proposed ANN models. In addition to the original inputs, two extra input factors were included in the cross-correlation analysis: water level variation (*W_r_*) and the frequency of water level change (*W_f_*) for consecutive records in a time interval (e.g., 1 h). The definitions of *W_r_* and *W_f_* are shown below:(5)$$W_{r,i}=X_{obs,i}-X_{obs,i-1}$$
(6)$$W_{f,i}=W_{r,i}-W_{r,\;i-1}$$

Mathematically, *W_r_* and *W_f_* represent the first- and second-order differences of the water level sequence at a target location, respectively. Physically, *W_r_* and *W_f_* are the velocity and acceleration of the change water level, respectively. Zhong et al. [40] found that considering the first- and second-order differences in the water level sequence can improve the forecasting accuracy of ANN models. However, they used this information to identify the level of data fluctuations and then applied the Kalman filter algorithm for local optimization. Details of the selected factors were not discussed in their study. In contrast, this study is the first attempt to apply these variables as input factors for the proposed ANN models to perform water level forecasting. Details of the analysis are described in the Results and Discussion section.

## 4. Results and Discussion

The ANN models were integrated into sensors, and their performance was evaluated in real cases. The discussion of the results is divided into three parts: (1) data preparation, (2) development, and (3) application. The research flowchart of this process is shown in Figure 5.

### 4.1. Data Preparation

#### 4.1.1. Generation of Synthetic Rainfall Data for Different Return Periods

Data quality and quantity are important for the accuracy of data-driven models (e.g., ANN models). The well-calibrated data from 2012 to 2017, including rainfall, flow discharge, and water stage, from the experimental watershed in the Yilan River Basin (Figure 1), were adopted. Following the research flowchart shown in Figure 5, a frequency analysis using rainfall data from five rainfall stations was conducted, and the magnitude of extreme events was related to the corresponding frequency of occurrence through the use of a probability distribution. To find the magnitude associated with a certain return period, the standard frequency factor method [41] is used, as follows:(7)$$X_{T}=\bar{X}+Ks$$
where $X_{T}$ is the calculated rainfall value in a certain period, $\bar{X}$ is the mean rainfall value from historical data, K is the frequency factor, and s is the standard deviation of the historical data. K was selected for 2-, 5-, 10-, 25-, 50-, and 100-year return period events based on the Pearson III distribution [42]. The calculated cumulative rainfall for a 24-h rainfall duration is shown in Table 1. In terms of topography, the upper parts of the Yilan River Basin have mountain topography and steep slopes. There are wide flood plains from the lower reaches to the Pacific Ocean. The stations YR_R2 and YR_R4 are located in mountain and floodplain areas, respectively. As a result, YR_R2 and YR_R4 have the greatest and fewest values among all stations. The rainfall values are consistent with the variations in topography.

Hydrographs specify the precipitation depth in 24 successive time intervals of 1 h duration over a total of 24 h. Such hydrographs are necessary inputs to hydrological and hydraulic models to generate flow discharge and water level data in the Yilan River Basin for different return periods. The details associated with hydrological and hydraulic models are discussed in the next section. The annual 24-h maximum rainfall value at each station from 2012 to 2017 was selected, and the average contribution (%) of each hour to the 24-h duration was calculated. These contributions were reordered in a time sequence with the maximum contribution occurring at the center of the 24-h duration and the remaining contributions arranged in descending order alternatively to the right and to the left of the center value to form a hyetrograph. The results are shown in Figure 6.

#### 4.1.2. Generation of Synthetic Water Level and Flow Discharge Data

The abovementioned synthetic rainfall data were used with the hydrological model Hydrologic Modeling System (HEC-HMS) and the hydraulic model River Analysis System (HEC-RAS) to generate discharge and water level data for five return periods. HEC-HMS simulates rainfall runoff processes at the watershed level and includes different components, such as the runoff volume, baseflow, and channel flow. For more details, please refer to the Technical Reference Manual [43] and User’s Manual [44]. In this study, an initial loss and a constant loss rate were subtracted from the precipitation depth, and the remaining depth was referred to as precipitation excess. Thereafter, excess precipitation was transformed to direct surface runoff through the SCS unit hydrograph (SCS-UH) method.

The total flow at YR_Q1 in the Yilan River Basin (Figure 1) was the sum of the direct runoff plus the base flow, and the base flow was obtained from an initial value multiplied by an exponential decay constant. The calculated total flow from the HEC-HMS model was then used as the upstream boundary condition for the downstream HEC-RAS model to calculate the variation in the water level downstream. To validate the performance of the HEC-HMS model, Typhoons Soulik (2012), Dujuan (2015), and Megi (2016) were considered. Table 2 shows the comparisons between the observations and simulations. Differences in peak discharge and time to peak discharge were below 15% and less than 2 h, respectively. Since the results met the relevant performance requirements, the calibrated model was then used to generate synthetic discharge values at YR_Q1 for further analysis. The inflow results associated with different return periods are shown in Figure 7.

There are four water level stations downstream of YR_Q1 (Figure 1). Among them, data from YR_S1, YR_S2, and YR_S3 were used as input data to train the ANN model. The HEC-RAS model was used to estimate water level variations at the abovementioned locations, and YR_S4 was used as the downstream boundary. HEC-RAS is a 1D river hydraulic model based on the Saint-Venant equations. These equations are approximated with the implicit Preissmann scheme and solved numerically using the Newton–Raphson iteration approach [45]. The downstream boundary condition was assumed to be the observed water levels during Typhoon Dujuan (2015). In comparison with observations, the performance was evaluated at YR_S2 in terms of temporal variations in water level, and the results are shown in Figure 8. The results demonstrate that differences in peak water level and time to peak water level were below 10% and zero hours, respectively. In conclusion, both the hydrological and hydraulic results confirm that the model parameters were well-tuned to generate the data needed for the development of the ANN model in the next section.

### 4.2. ANN Model Development

#### 4.2.1. Correlation Analysis and Input Factor Selection

As described in Section 3.2, precipitation from YR_R1 to YR_R5, flow discharge at YR_Q1, and water levels from YR_S1 to YR_S3 were assumed to be correlated with the water level output at YR_S2. A correlation analysis between the targeted water level at the present time and the abovementioned variables from previous periods was performed using Equation (4). This model was considered the ANN_0 model. To include physical features such as the velocity and acceleration of water level variations, a correlation analysis between the first-order and second-order differences of the output and its values from previous periods was conducted. In this way, two more models were developed, named the ANN_1 and ANN_2 models. All of the correlation results are shown in Table 3. The variables with the highest correlation results with those in previous periods were selected as the model inputs. For example, for the ANN_0 model, the highest correlation results are 0.795 and 0.917 for YR_R1 at t−4 h and YR_S1 at t−1 h, respectively. As a result, four variables of YR_R1 and one variable of YR_S1 were selected for the ANN_0 model. However, in some cases, such as YR_R1 in the ANN_1 model, the correlation results 7 and 8 h earlier were 0.601 and 0.608, respectively. These values were different at three decimal digits. Other rainfall-related input factors, such as YR_R2, YR_R3, YR_R4, and YR_R5, were the variables 7 h earlier. To be consistent with other rainfall input factors and to increase computational efficiency, the variables 7 h earlier were selected for YR_R1. In addition, few variables were included, so the efficiency was increased considering the limited computing resources of the sensors. As a result, there were a total of 26, 46, and 46 input factors for the ANN_0, ANN_1, and ANN_2 models, respectively, and details of the input factor selection are listed in Table 4.

#### 4.2.2. Model Training and Testing

Random sampling was employed to split the input data from Section 4.1 into training and testing sets at an 80–20% ratio. For more details in terms of random sampling, a dataset included the input factors listed in Table 4. For example, there were 46 data needed in a dataset to train the model and produce water levels at t + 1, t + 2, and t + 3. Shown in Table 4, all data in the dataset were in a sequential order. The amount of dataset was depended on the data availability during the training process. For example, if a 24-h was available, there were 15 datasets available for training process. At each training, observed or synthetic water levels at t + 1, t + 2, t + 3 were used for performance evaluation. Two extra experiments were conducted. One involved using data from 2-, 5-, 10-, 25-, and 50-yr return periods for training and data from the maximum 100-yr return period for testing. Another involved using data from 5-, 10-, 25-, 50-, and 100-yr return periods for training and data from the minimum 2-yr return period for testing. The number of training data was fully prepared and the number of the data was constant, therefore, there was only one epoch done during this training process. The purpose of these experiments was to assess the ANN models and their forecasting capability beyond the training data range. The RMSE index (Equation (3)) was used to evaluate the model performance. The forecasting results with 1-, 2-, and 3-h lead times at location YR_R2 are shown in Table 5.

The results demonstrated that the worst performance among the three models for all three lead times was from the ANN_0 model; its RMSE results were 0.3421 m and 0.7743 m for 1- and 3-h lead times, respectively. The performance of the other two tests for all three models was comparable to that in the cross-validation test. This finding confirmed that the proposed ANN models have the capability to forecast data beyond the testing data range. In comparison, the ANN_1 and ANN_2 models yielded RMSEs that were all below 0.2 m. The ANN_2 model displayed better performance for the results with 2- and 3-h lead times than ANN_1. However, the performance of these two models deteriorated when the forecasting lead time was increased. According to the abovementioned results, the following tests were continuously implemented using the ANN_1 and ANN_2 models.

To test the performance of the proposed ANN models for operational purposes, three historical typhoon events, namely, Yutu (2018), Mangkhut (2018), and Maria (2018), were considered. Three typhoons were split into two typhoons for training and one typhoon for testing. The results are demonstrated in Table 6, and the naming convention is based on the names used in the test case. The performance of both models was fairly consistent. All RMSEs were below or close to 0.1 m regardless of the lead time. According to the individual results, the ANN_2 model performed slightly better than the ANN_1 model. A temporal comparison between observations and the simulations of both models is shown in Figure 9. The results demonstrated that both models agree fairly well with the observations for all three typhoons. It was interesting to find that performance did not deteriorate when the lead time was increased. In contrast, both models produced worse results with a 1-h lead time when the peak water level occurred during Typhoon Yutu in comparison with those results for 2- and 3-h lead times.

Additionally, the calculation times of the models were compared for different hardware configurations. The comparison was conducted with a hardware configuration that included an AMD Ryzen 94900 central processing unit, an NVIDIA GeForce RTX 2060 (PC) and an RPi 3B+ (the sensor mentioned in Section 3.1), among other components. Using Typhoon Yutu with the ANN_2 model as an example, the calculation time was 5 min if the model was run on the PC and 30 min if it was run on the sensor. The results for the PC and the sensor were consistent, but the calculation time when the model ran on the sensor was 6 times slower than when it ran on the PC. Therefore, calculation time is an issue that must be addressed in future investigations if the sensor is installed on site for real operation. In conclusion, all the results above confirmed the capability of the proposed ANN models to forecast water levels with up to a 3-h lead time. According to the comparison of model performance, the ANN_2 model can be continually applied for further applications and will be discussed in the next section.

### 4.3. Applications

The proposed ANN_2 model and integrated sensor were then applied to three other canals and rivers for real-world tests. The details of the geographic locations of these three study areas can be found in Figure 1. The same input factors as in Table 4 were selected, but the number of gages varied according to the number of gages installed in the study areas. For example, there is only one rainfall station near the Shimen Canal; therefore, the number of input factors decreased from 46 to 17. The list of the input factors for these three study areas is shown in Table 7. In addition to the RMSE shown in Equation (3), the coefficient of determination (*R*^2^) described below was used to evaluate system performance in these real-world cases.
(8)$$\begin{array}{c} R^{2}=1-\frac{SS_{res}}{SS_{tot}}\\ SS_{res}=\sum {\left(P_{i}-O_{i}\right)}^{2}\\ SS_{tot}=\sum {\left(P_{i}-\bar{O}\right)}^{2} \end{array}$$
where $O_{i}$ and $\bar{O}$ are the hourly observations and mean of the observations, respectively, and $P_{i}$ is the prediction. If the predictions exactly match the observations, $SS_{res}$ = 0 and $R^{2}$ = 1. A detailed discussion for each case is given below. In the applications, the sensor was continually receiving data. If the sensor collected new data, the model was retraining with newly collected data. Therefore, a new epoch was completed. This process was repeated until the end of the experiment.

#### 4.3.1. Shimen Canal

The distance between SC_R1 and SC_R3 is approximately 650 m and is shown in Figure 1. Three sensors were installed at SC_R1, SC_R2, and SC_R3. The experiment was performed on site during 24 June 2021, and 25 June 2021. There were a total of 200 observations collected by each sensor for an hour, and the average value was used for model input. Other data, such as discharge and precipitation data, were retrieved from local stations. The system started to produce water levels at SC_R2 with 1-, 2-, and 3-h lead times after the fifth hour of installation. The overall testing time was 35 h. The comparison between observations and simulations is shown in Figure 10. The errors, which were defined as the difference between observation and prediction, were within 0.075 m. The largest error of 0.075 m was found in Figure 10a with a 1-h lead time. One possible reason for the error was human interference. The canal is operated at a certain water level for the purpose of irrigation. This study did not include the factors of human operations, and the performance deteriorated once human inference was carried out. The RMSEs were between 0.02 and 0.03 m. According to the performance, the system was able to produce the variation in the water levels. However, because of manual operations such as gate control upstream of the canal, *R*^2^ was only in the range of 0.3 to 0.45. The total computation time needed to train the proposed ANN_2 model in the sensor and produce water level forecasts using the observations was 8 min.

#### 4.3.2. Toucian and Beinan Rivers

To avoid the impact of manual operations on system performance, the Toucian and Beinan Rivers were selected to test system performance. Unfortunately, the distance between the observed water surface and the installation of the sensor was beyond the range of the maximum measurement distance. Therefore, the following experiments were conducted by using the data retrieved from the Water Resources Agency directly. The evaluation period was from 3 June 2021, to 31 December 2021, for the Toucian River. Typhoon In-fa on July 24 and Typhoon Lupit on August 6 influenced this study area during this period. Figure 11 shows that the largest difference between observations and predictions at TR_S2 (see the location in Figure 1) was 1.3 m among all forecasts. This was the time when Typhoon In-fa had an impact on this area (solid circle in Figure 11). Similar to previous cases, the system started to produce forecasts the 5th hour after the experiment started. It was confirmed that having more data in the training process increased the accuracy of the ANN_2 model. The errors between observations and simulations were decreased to below 0.5 m for the second typhoon event (dashed circle in Figure 11) and thereafter. Finally, the overall R^2^ and RMSEs were approximately 0.98 and 0.045 m, respectively.

For the Beinan River, the evaluation period was from 1 July 2021, to 31 December 2021. Typhoon Lupit on August 6 and Typhoon Kompasu on October 14 influenced this study area during the period. Figure 12 shows that the greatest difference between the observations and predictions at BR_S2 (see the location in Figure 1) was below 1 m. The largest errors were found when the highest and lowest water levels were observed (solid and dashed circles in Figure 12), and extreme values such as these were not included in the training data. The overall R^2^ and RMSEs were approximately 0.98 and 0.08 m, respectively. In conclusion, based on the above experiments, the proposed ANN_2 model and integrated sensor show excellent potential to perform edge computing locally and generate water level forecasts for real-time operation. The forecasting accuracy was influenced if the water levels were beyond the values in the training data set. Furthermore, the performance of the proposed system decreased if human interference occurred in the study area.

## 5. Conclusions

In this study, an ANN-based model was integrated into a Raspberry Pi-based sensor to implement edge computing for hourly river water level forecasting. The ANN model is capable of forecasting the river level with a high level of precision by only using previously observed water level, rainfall information, and flow discharge as inputs without the need for other hydrological and meteorological parameters. Edge computing is a form of computing that is conducted on site or near a targeted location, thus minimizing the need for data to be processed at a remote data center and increasing the efficiency of the emergency response. This study is a first attempt to combine real-time customized sensors and ANN algorithms in practice. Based on historical measured data from the Yilan River in Yilan County, synthetic upstream rainfall and discharge data were generated for six different return periods using the Pearson III probabilistic model and the HEC-HMS hydrological model with synthetic inflow data. The downstream water level data were obtained with the aid of the HEC-RAS hydraulic model. Different input combinations, including first-order difference and second-order difference water levels, were investigated to enhance the precision of the ANN model. The results demonstrated that the proposed ANN model has the capability to precisely forecast the future fluctuations in the water level of rivers in a short time and with a small number of inputs. The model was then embedded into the customized sensor to forecast the water level over different time horizons up to 3 h in advance. Finally, a comprehensive comparison between forecasts and observations was performed for three other rivers and canals. The findings revealed that the proposed water level sensor with the ANN model exhibited a high level of performance when it was applied to real events. Therefore, the integration proposed in this study is very promising and could be incorporated into a new generation of flood warning systems to prevent and mitigate the impacts of floods in downstream areas. However, the time required to train the ANN model and the system to produce results is over 30 min. Due to the limited computing power of the sensor, the amount of data required to train the model during real-time operation while maintaining high forecasting accuracy to speed up the computing process needs to be investigated further. In addition, different options of microcontroller units with more computing power will also be considered in future studies. Finally, in the current study, only an on-site experiment was performed because of the limitation of devices. More on-site experiments must be performed, and input factors of the ANN model regarding human interference should be included in future studies to extend the system application scope.

## Figures and Tables

**Figure 1 sensors-22-08532-f001:**
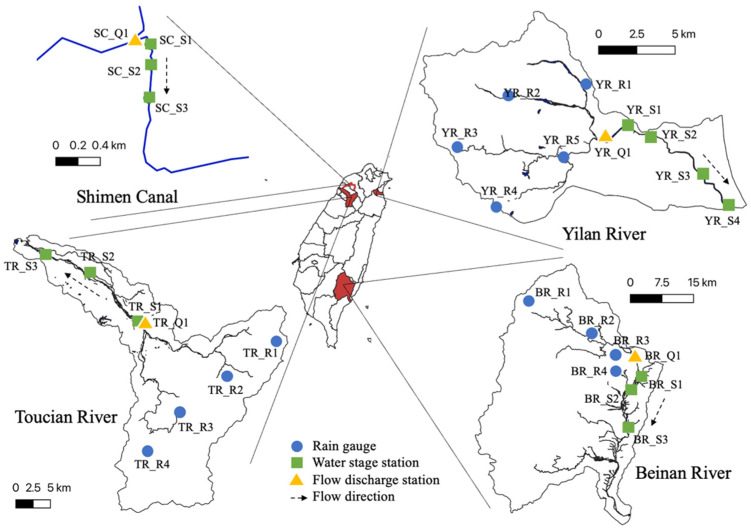
The four study areas are the Shimen canal in Taoyuan County (**upper left**), the Yilan River in Yilan County (**upper right**), the Beinan River in Taitung County (**bottom right**), and the Toucian River in Hsinchu County (**bottom left**), Taiwan.

**Figure 2 sensors-22-08532-f002:**
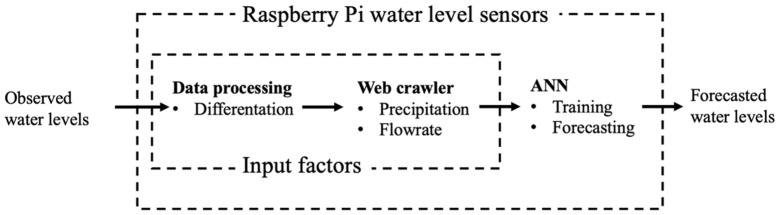
System data processing flowchart.

**Figure 3 sensors-22-08532-f003:**
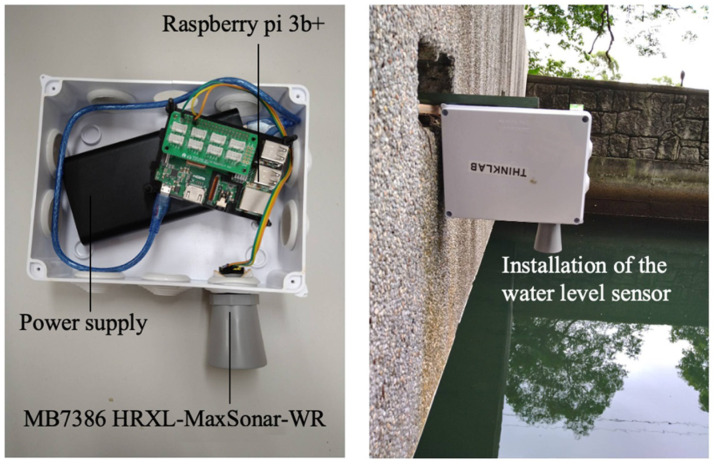
Details of the RPi-based ultrasonic water level sensor (**left**) and the installation of the sensor (**right**).

**Figure 4 sensors-22-08532-f004:**
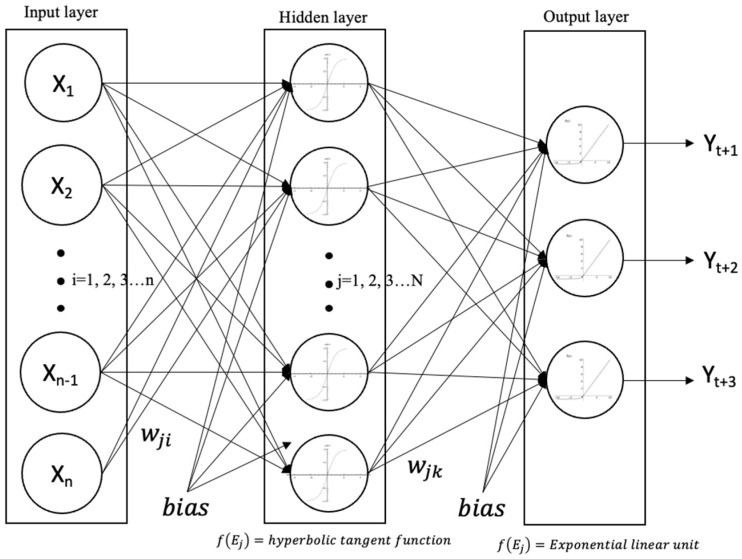
A three-layer ANN model and its data processing flowchart.

**Figure 5 sensors-22-08532-f005:**
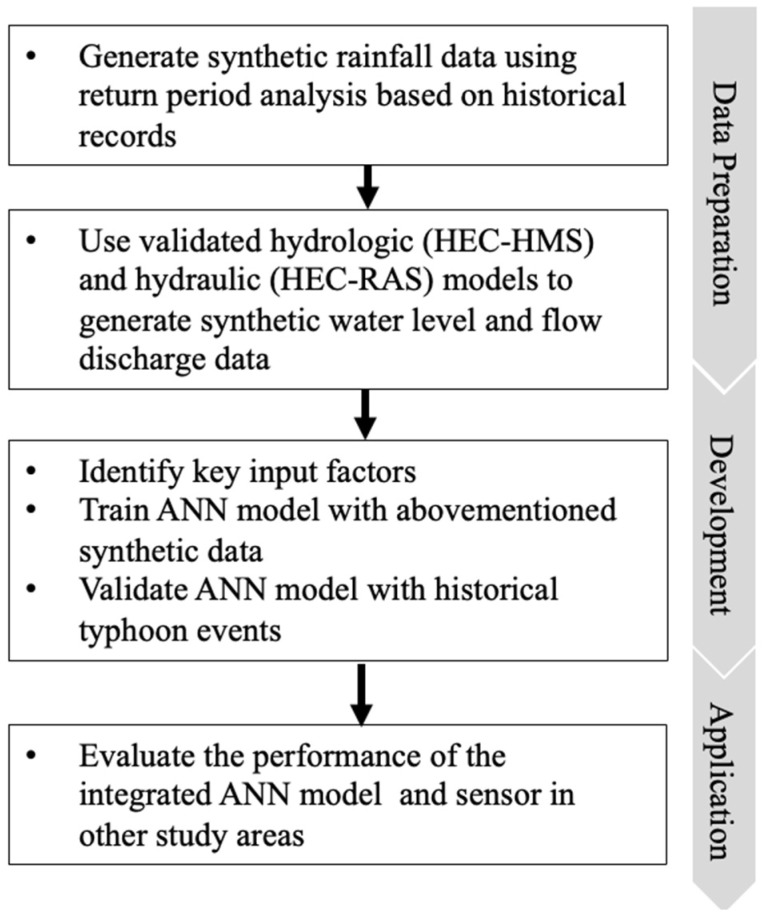
Research flowchart of the proposed ANN models.

**Figure 6 sensors-22-08532-f006:**
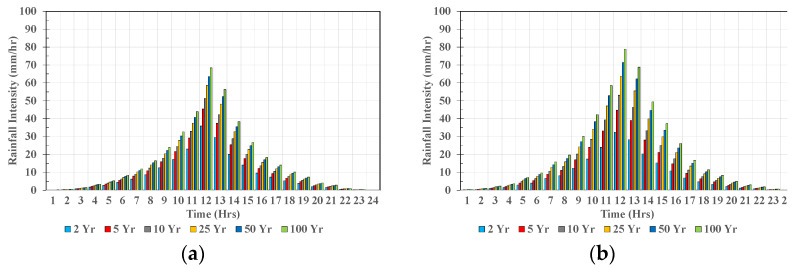
(**a**–**e**) The hourly rainfall distribution for different return periods in the Yilan River basin at YR_R1-YR_R5.

**Figure 7 sensors-22-08532-f007:**
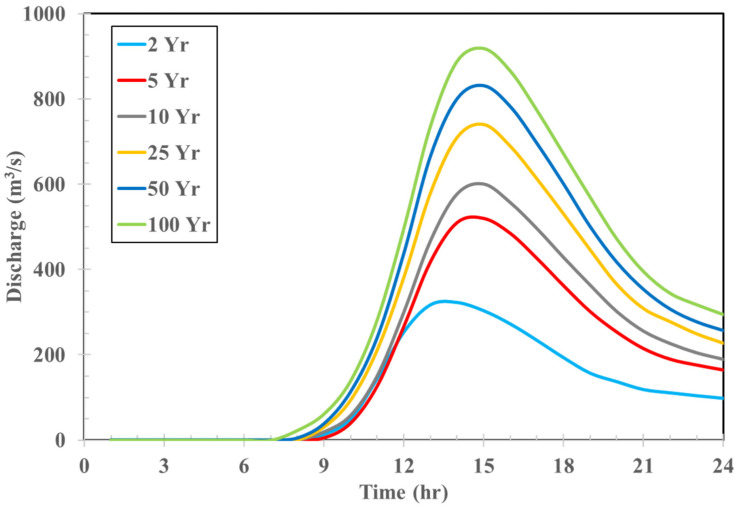
Synthetic inflow hydrograph at YR_Q1 in the Yilan River basin for different return periods.

**Figure 8 sensors-22-08532-f008:**
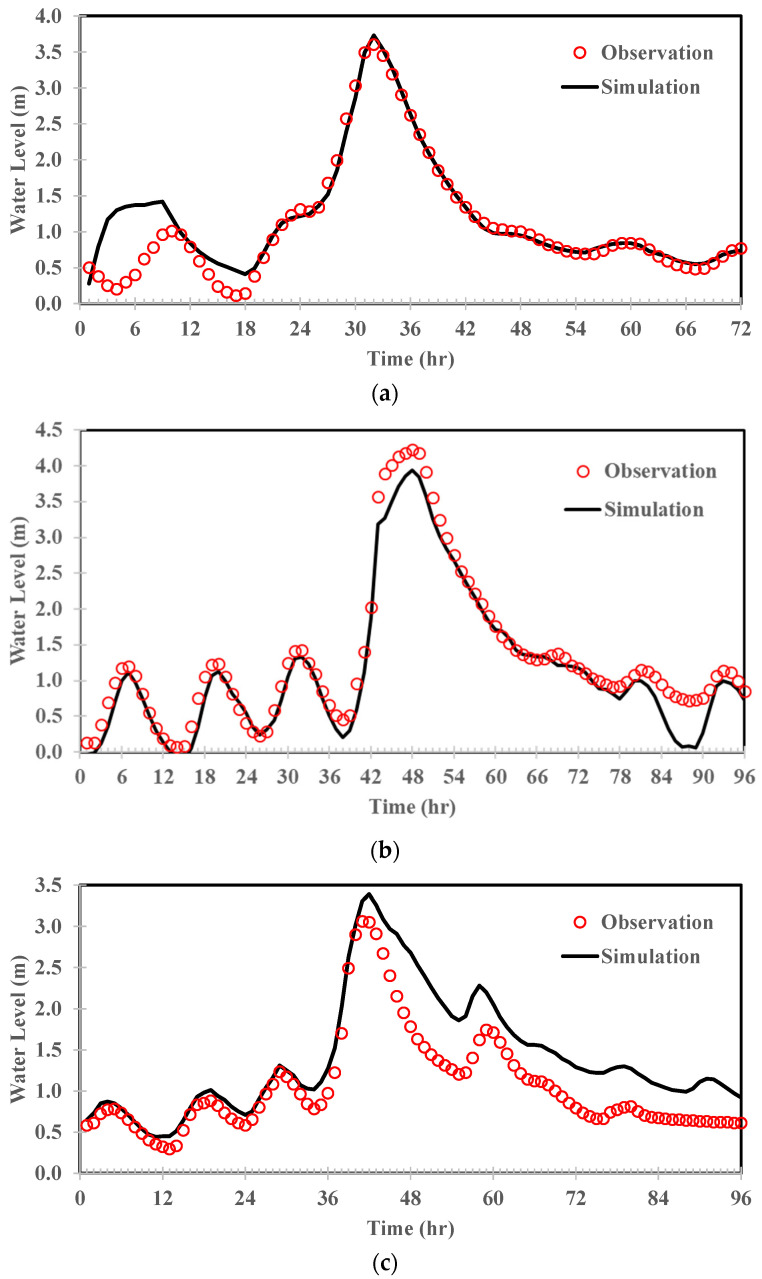
Comparison of the simulated and observed water levels at YR_R2 for (**a**) Typhoon Soulik; (**b**) Typhoon Dujuan; and (**c**) Typhoon Megi.

**Figure 9 sensors-22-08532-f009:**
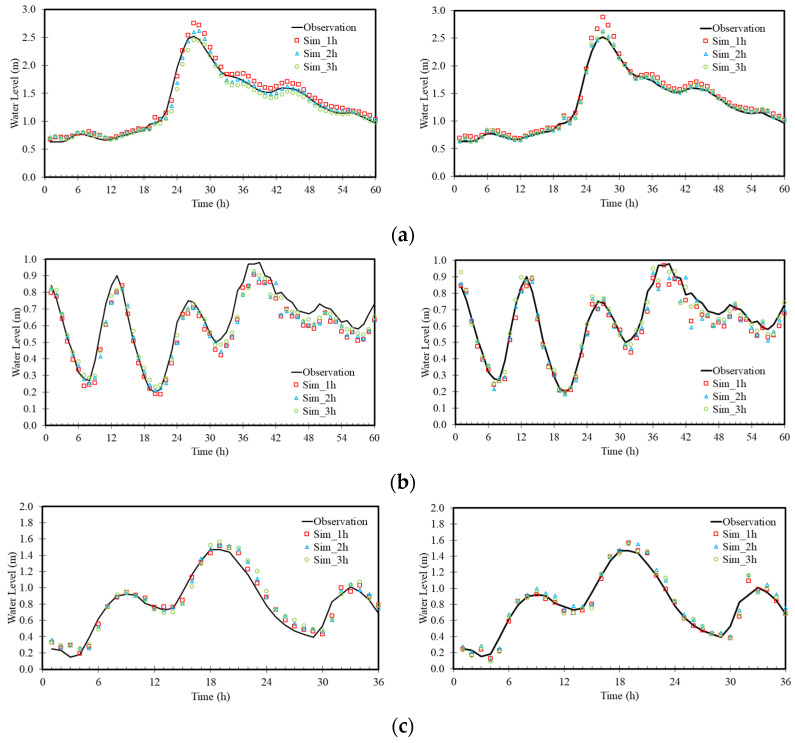
Comparison of simulated and observed water levels at YR_R2 for the ANN_1 model (**left**) and ANN_2 model (**right**) at Yutu (**a**), Mangkut (**b**), and Maria (**c**).

**Figure 10 sensors-22-08532-f010:**
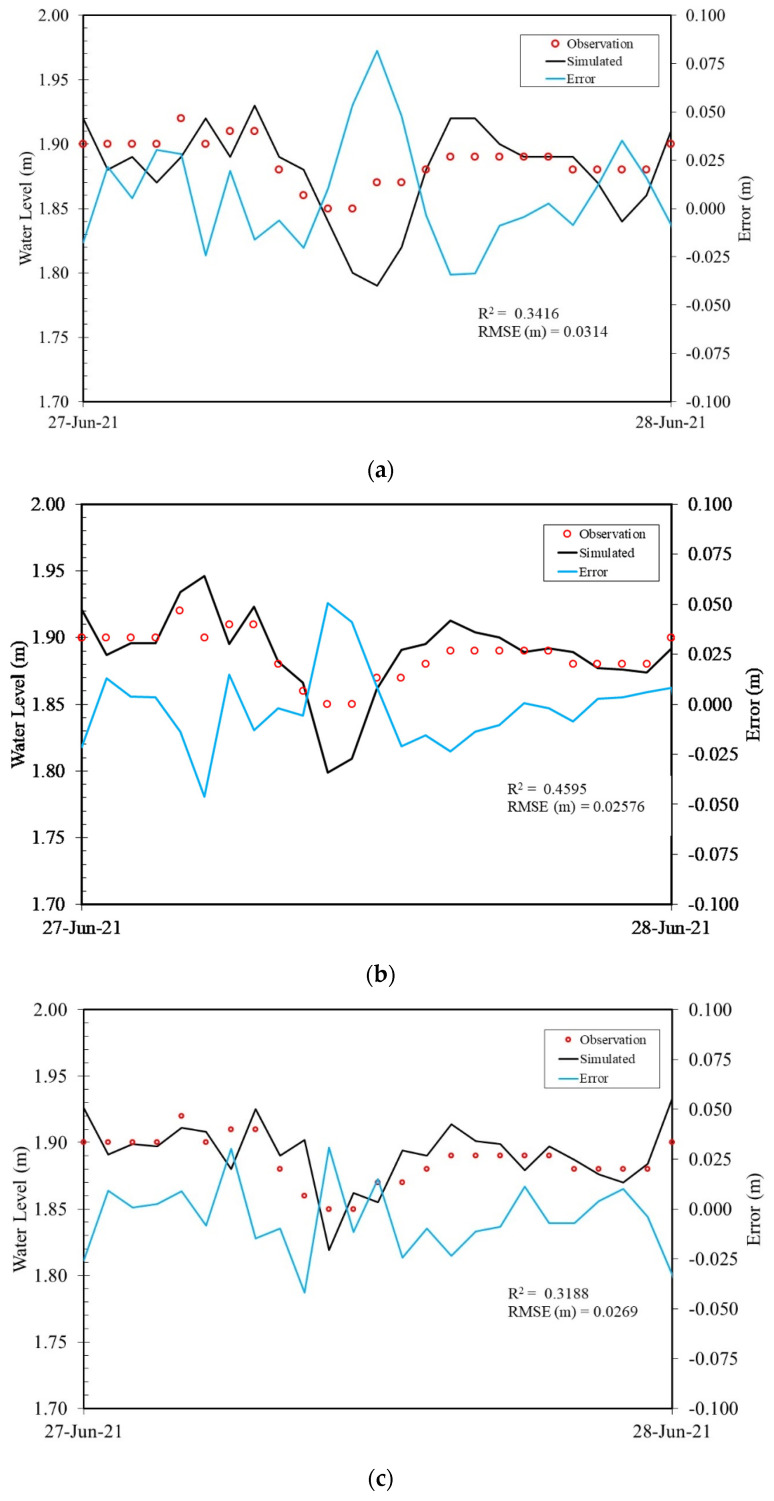
(**a**–**c**) Comparison of simulation and observed water levels at SC_S2 of Shimen canal for lead times t + 1 h, t + 2, and t + 3 h, respectively.

**Figure 11 sensors-22-08532-f011:**
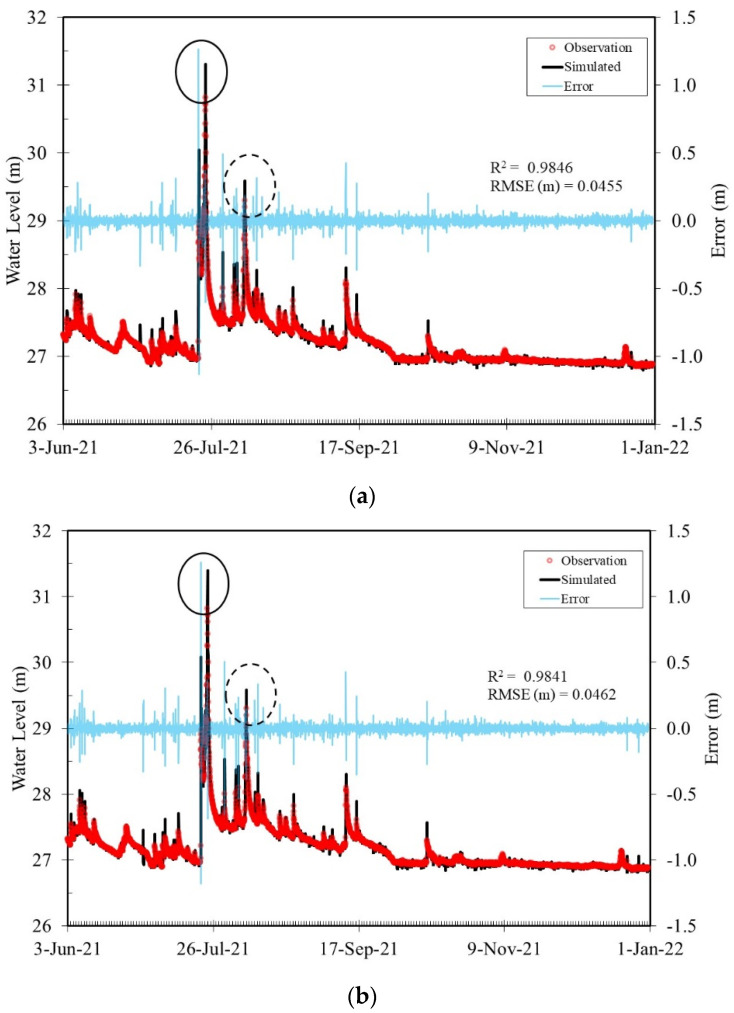
(**a**–**c**) Comparison of simulation and observed water levels at TR_S2 of the Touciaan River for lead times t + 1 h, t + 2, and t + 3 h, respectively.

**Figure 12 sensors-22-08532-f012:**
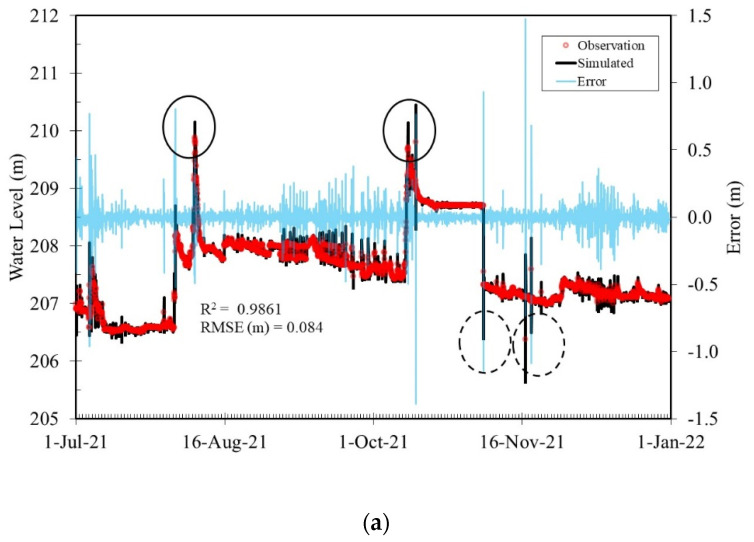
(**a**–**c**) Comparison of simulation and observed water levels at BR_S2 of the Beinan River for lead times t + 1 h, t + 2, and t + 3 h, respectively.

**Table 1 sensors-22-08532-t001:** Calculated 24-h rainfall values at five stations for different return periods.

Station ID	24-h Duration Rainfall for Different Return Periods in mm					
	2-yr	5-yr	10-yr	25-yr	50-yr	100-yr
YR_R1	208.96	264.35	298.96	340.15	369.35	397.30
YR_R2	205.90	283.71	336.73	403.20	451.81	499.36
YR_R3	191.88	256.75	305.44	369.67	415.85	466.46
YR_R4	178.11	220.21	242.81	267.33	283.37	298.00
YR_R5	263.98	332.60	369.57	408.09	433.36	456.17

**Table 2 sensors-22-08532-t002:** The validated performance of HEC-HMS for three typhoons.

Event	Soulik (2012)	Dujuan (2015)	Megi (2016)
Difference in peak discharge (%)	14.8	3.83	14.1
Difference in time to peak discharge (hour)	1	2	0

**Table 3 sensors-22-08532-t003:** Correlation matrix for output and input variables.

Model	Lag (Hour)	YR_R1	YR_R2	YR_R3	YR_R4	YR_R5	YR_Q1	YR_S1	YR_S2	YR_S3
ANN_0	1	0.586	0.595	0.534	0.538	0.576	0.863	0.917	0.923	0.887
	2	0.712	0.721	0.652	0.662	0.701	0.784	0.788	0.795	0.777
	3	0.782	0.791	0.710	0.725	0.767	0.661	0.625	0.639	0.652
	4	0.795	0.807	0.716	0.734	0.780	0.507	0.440	0.465	0.519
	5	0.762	0.775	0.678	0.697	0.742	0.336	0.243	0.278	0.376
ANN_1	1	−0.278	−0.281	−0.297	−0.306	−0.274	0.394	0.488	0.680	0.471
	2	−0.037	−0.046	−0.044	−0.045	−0.031	0.555	0.601	0.232	0.494
	3	0.177	0.171	0.173	0.183	0.182	0.645	0.643	0.054	0.462
	4	0.348	0.341	0.333	0.353	0.349	0.664	0.640	0.070	0.445
	5	0.473	0.467	0.442	0.470	0.472	0.624	0.612	0.138	0.460
	6	0.557	0.533	0.510	0.549	0.552	-	-	-	-
	7	0.601	0.595	0.542	0.585	0.589	-	-	-	-
	8	0.608	0.604	0.540	0.582	0.595	-	-	-	-
	9	0.570	0.571	0.499	0.539	0.559	-	-	-	-
ANN_2	1	−0.260	−0.264	−0.283	−0.291	−0.257	0.400	0.488	0.680	0.471
	2	−0.013	−0.022	−0.026	−0.024	−0.008	0.562	0.601	0.231	0.494
	3	0.205	0.198	0.195	0.207	0.209	0.651	0.643	0.054	0.462
	4	0.376	0.369	0.357	0.379	0.377	0.665	0.640	0.070	0.445
	5	0.499	0.494	0.465	0.498	0.497	0.618	0.612	0.138	0.460
	6	0.577	0.573	0.531	0.573	0.568	-	-	-	-
	7	0.609	0.604	0.557	0.599	0.595	-	-	-	-
	8	0.604	0.604	0.546	0.589	0.592	-	-	-	-
	9	0.556	0.556	0.493	0.527	0.541	-	-	-	-

**Table 4 sensors-22-08532-t004:** ANN models and their input factors.

Model Name	Input Combination	No. of Factors
ANN_0	YR_R1(t−1…,t−4), YR_R2(t−1…,t−4), YR_R3(t−1…,t−4), YR_R4(t−1…,t−4), YR_R5(t−1…,t−4), YR_Q1(t−1), YR_S1(t−1), YR_S2(t…, t−2), YR_S3(t−1)	26
ANN_1	YR_R1(t−1…,t−7), YR_R2(t−1…,t−7), YR_R3(t−1…,t−7), YR_R4(t−1…,t−7), YR_R5(t−1…,t−7), YR_Q1(t−1…,t−4), YR_S1(t−1…,t−3), YR_S2(t, t−1), YR_S3(t−1, t−2)	46
ANN_2	YR_R1(t−1…,t−7), YR_R2(t−1…,t−7), YR_R3(t−1…,t−7), YR_R4(t−1…,t−7), YR_R5(t−1…,t−7), YR_Q1(t−1…,t−4), YR_S1(t−1…,t−3), YR_S2(t, t−1), YR_S3(t−1, t−2)	46

**Table 5 sensors-22-08532-t005:** RMSEs of different ANN models for the training and testing processes.

Model	Lead Time (Hours)	Cross-Validation RMSE (m)	Test for 100-yr Event RMSE (m)	Test for 2-yr Event RMSE (m)
ANN_0	1	0.3421	0.4532	0.5935
	2	0.4142	0.3192	0.5431
	3	0.7743	0.5878	0.6054
ANN_1	1	0.0553	0.1212	0.1065
	2	0.1085	0.1962	0.1129
	3	0.1655	0.2162	0.0987
ANN_2	1	0.1229	0.1204	0.1155
	2	0.0678	0.1029	0.1135
	3	0.0778	0.1253	0.1286

**Table 6 sensors-22-08532-t006:** Comparison of RMSEs for different ANN models and historical typhoon events.

Model	Lead Time (Hour)	Typhoon Yutu (2018) RMSE (m)	Typhoon Mangkhut (2018) RMSE (m)	Typhoon Maria (2018) RMSE (m)
ANN_1	1	0.0954	0.0751	0.0756
	2	0.0759	0.0729	0.0931
	3	0.1000	0.0611	0.1069
ANN_2	1	0.0977	0.0591	0.0742
	2	0.0508	0.0611	0.0893
	3	0.0530	0.0475	0.0912

**Table 7 sensors-22-08532-t007:** ANN models and their input factors for various applications.

Model Name	Input Combination	No. of Factors
Shimen Canal model	SC_R1(t−1…,t−7), SC_Q1(t−1…,t−4), SC_S1(t−1…,t−3), SC_S2(t, t−1), SC_S3(t−1, t−2)	17
Toucian River model	TR_R1(t−1…,t−7), TR_R2(t−1…,t−7), TR_R3(t−1…,t−7), TR_R4(t−1…,t−7), TR_Q1(t−1…,t−4), TR_S1(t−1…,t−3), TR_S2(t, t−1), TR_S3(t−1, t−2)	39
Beinan River model	BR_R1(t−1…,t−7), BR_R2(t−1…,t−7), BR_R3(t−1…,t−7), BR_R4(t−1…,t−7), BR_Q1(t−1…,t−4), BR_S1(t−1…,t−3), BR_S2(t, t−1), BR_S3(t−1, t−2)	39

## Data Availability

The experimental watershed data can be accessed in [23], and the data that support the findings of this study are available from the corresponding author, T.-H.Y., upon reasonable request.

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
