# Peer review of "Development of an Artificial Neural Network Algorithm Embedded in an On-Site Sensor for Water Level Forecasting"

_sensors, 2022, doi:10.3390/s22218532_

Round 1

Reviewer 1 Report

See attached file.

Author Response

Dear Reviewer,

Please see the response in the attached file.

The authors appreciate your comments and recommendations.

It makes this manuscript much better than befpre.

Reviewer 2 Report

1. The discussion of related research work is relatively weak, which needs to be further improved to fully reflect the advantages of this technology. The merits of the proposed approach need to be clarified. The contribution of the paper needs to be stated more clearly.

2. Are there any other solutions in this field? If there are any other typical methods, you can try to compare them.

3. This paper is of certain significance to the prevention of flood disasters in theory, but it is suggested to explain the rigor of the experiment in this paper.

4. The conclusion should put forward some concrete ideas for future work.

5. The number of references should be increased as much as possible in recent years and classics.

Author Response

(The authors gave the same response as above.)

Reviewer 3 Report

This paper applies the ANN-based model to forecast hourly river water level, and implements edge computing with Raspberry Pi-based sensor.

Some detailed comments are listed below:

1.The meaning of HEC-HMS is not given in line 308.

2.Line 370, the author said: “The variables with the highest correlation results with those in previous periods were selected as the model inputs”, however, in table4, YR_R1 has 4 parameters, but YR_S1 only has 1 parameter, please give the detailed standard for picking up proper parameters.

3.Line 370, the author said: “However, in some cases, such as YR_R1 in the ANN_1 model, the correlation results 7 and 8 hours earlier were 0.601 and 0.608, respectively. These values were almost identical, and therefore, the variable 7 hours earlier was selected to be consistent with other variables that were 7 hours earlier”, please give the explanation for this exception instead of telling us the results.

4.In figure 10, it’s necessary to give some explanation for the error in the middle of (a) and (b).

Author Response

(The authors gave the same response as above.)

Round 2

Reviewer 1 Report

All of my comments were properly addressed. Well done! To me, the paper is now ready for publication.

Author Response

Dear Reviewer,

The authors appreciate your comments and recommendations.

It makes this manuscript much better than before.

Reviewer 3 Report

I agree to the author's modification.

Author Response

(The authors gave the same response as above.)
